

# The CAMELS data set: catchment attributes and meteorology for large-sample studies

Nans Addor[1], Andrew J. Newman[1], Naoki Mizukami[1], Martyn P. Clark[1]

[1] Research Applications Laboratory, National Center for Atmospheric Research, Boulder, USA

*Correspondence to*: Nans Addor (naddor@ucar.edu)

**Abstract.** We present a new data set of attributes for 671 catchments in the contiguous USA (CONUS). This complements the daily hydrometeorological time series provided by Newman et al. (2015b) and opens new opportunities to explore how the interplay between landscape attributes shapes hydrological processes and catchment behavior. To produce this extension,

we synthesized diverse and complementary data sets to describe topography, climate, hydrology, soil and vegetation characteristics at the catchment scale. The spatial variations among basins over the CONUS are discussed and compared using a series of maps. The large number of catchments, combined with the diversity of their geophysical characteristics, makes this new data well suited for large-sample studies and comparative hydrology. An essential feature, that differentiates this data set from similar ones, is that it both provides quantitative estimates of diverse catchment attributes, and involves

assessments of the limitations of the data and methods used to compute those attributes. This data set will be publicly available and we encourage the community to further extend it. The hydrometeorological time series provided by Newman et al. (2015b) together with the catchment attributes introduced in this paper constitute the CAMELS data set: Catchment Attributes and MEteorology for Large-sample Studies.

## 1 Introduction

Catchment attributes are descriptors of the landscape. Their interplay shapes catchment behavior by influencing how catchments store and transfer water. Because they are designed to capture and synthesize the multi-faceted composition of catchments, catchment attributes necessarily cover a wide range of features, such as the catchment climate, hydrology, land cover, soil, geology, topography and river network. Over the last decades, catchment attributes have been developed in a variety of ways and are building blocks of countless hydrological studies.

A fruitful research direction involves studies that explore interrelationships among catchment attributes. Key examples include how the interaction of climate and topography influences vegetation productivity (Voepel et al., 2011), how aridity affects the angle of stream intersections, thereby constraining the shape of the river network (Seybold et al., 2017) or the extent to which land cover influences annual streamflow (Oudin et al., 2008) or evapotranspiration (Zhang et al., 2001).



Catchment attributes are also a standard way to characterize catchment (dis)similarities and are consistently employed to develop catchment classifications (e.g., McDonnell and Woods, 2004; Wagener et al., 2007; Sawicz et al., 2011; Berghuijs et al., 2014). Furthermore, there have been considerable efforts to use catchment attributes to reflect the structure of the landscape in models. One approach is to infer hydrological model parameter values from catchment attributes (Seibert, 1999;

Hundecha et al., 2008; Samaniego et al., 2010; Hrachowitz et al., 2013), with the parallel objectives of accounting for landscape characteristics in an explicit way (not only implicitly by calibration), and of implementing hydrological models in ungauged basins. Another approach is to base not only parameter values, but also the choice of model structure on catchment attributes (Clark et al., 2011; McMillan et al., 2011; Fenicia et al., 2014). Both approaches provide guidance on how to deal with geophysical characteristics that vary dramatically within the model domain, for instance in the context of continental

scale modeling.

Although catchment attributes are routinely used when working with a handful of catchments, there is a growing recognition that a large sample of catchments can provide insights that cannot be gained from a small sample (Gupta et al., 2014). Large-sample data sets enable us to concentrate on catchment similarities, and on the formulation of conclusions that are valid for a

large number of (gauged and ungauged) catchments. Individual catchments can then be considered to be part of a continuum of catchment attributes, which vary in space along several gradients (such as aridity or soil depth). Working with a large number of catchments enables us to progressively move along different gradients, and to better disentangle the effects of catchment attributes on catchment behavior. This is particularly useful for comparative hydrology, i.e., to identify how similarities and differences between locations influence eco-hydrological processes (Falkenmark and Chapman, 1989; Troch

et al., 2009; Thompson et al., 2011; Harman and Troch, 2014). Further, large-sample hydrology opens new opportunities for data analysis, and for instance makes it possible to explore interrelationships between catchment attributes on the basis of their spatial patterns, as exemplified later in this study using map comparisons.

Several data sets of catchment attributes for large-sample hydrology now exist (see the review by Gupta et al., 2014). The

large-sample data set introduced in this paper is an extension of the Newman et al. (2015b) data set, referred to as N15 in continuation. N15 covers 671 catchment in the contiguous USA (CONUS), for which it provides daily meteorological forcing from three data sets, Daymet (Thornton et al., 2012), Maurer (Maurer et al., 2002) and NLDAS (Xia et al., 2012), as well as daily streamflow measurements from the United States Geological Survey (USGS). Here we cover the same catchments and provide additional quantitative estimates of a wide range of catchment attributes. We named this extended

N15 data set the CAMELS data set, which stands for Catchment Attributes and MEteorology for Large-sample Studies.

Section 2 explains the motivations to extend the N15 data set. Sections 3 to 7 present five classes of basin characteristics: topographic characteristics, climate indices, hydrological signatures, and soil and vegetation characteristics, respectively. These 5 sections are organized using the following structure. We first provide some research background on this class of





basin characteristics, introduce the attributes we selected and explain the reasons behind their selection. Since these attributes are well established, we briefly introduce them in the main text and provide further details in tables, which contain units and abbreviations for the attributes, as well as references to the equations and data sets used to for their computation. We follow by discussing the spatial variations of these attributes across the CONUS and by assessing their main limitations.

Section 8 compares the CAMELS data set to the MOPEX data set (Duan et al., 2006; Schaake et al., 2006), another large-sample hydrometorological data set for the CONUS. Section 9 discusses the online availability of the CAMELS data set and possible future extensions. Conclusions are presented in Section 10.

## 2. Motivations to extend the Newman et al. (2015b) data set

In creating the CAMELS data set we seek to achieve the following objectives:

1. Define a wide range of catchment attributes: We compiled complementary catchment attributes from diverse data sources and synthesized them into a single coherent data set. These attributes have been available separately for some time, but comprehensive multivariate catchment scale assessments have so far been difficult, typically because disparate data sets have different spatial configurations, are stored in different archives, and use different data formats. By creating catchment scale estimates of these attributes, we simplify the assessment of their
interrelationships.

2. Summarize meteorological forcing and discharge time series: We derived climate indices and hydrological signatures using the daily time series from N15. We selected climate indices and hydrological signatures that reduce the dimensionality of the hydroclimatic data sets, while preserving most of their information content. In other words, daily time series are rich in information, but summarizing this information makes catchment comparison
easier.

3. Characterize soil and vegetation: We included data sets not used in N15 in order to describe the soil and vegetation of each catchment. Those characteristics are essential to better understand and simulate key catchment processes, such as infiltration or evapotranspiration. We extracted soil and vegetation attributes commonly used to explore catchment behavior and to estimate parameters of hydrological and land surface models.

4. Define limitations in catchment attributes: Our intention is not only to provide quantitative estimates of diverse catchment characteristics, but also to explore and discuss limitations of those estimates. Catchment attributes are uncertain for different reasons, so we are providing metadata of different kinds (e.g., the proportion of missing values in discharge time series or estimates of the same quantity using different data set). Our aims are i) to contribute to raise awareness on uncertainties in geophysical attributes, which are frequently considered in a purely
deterministic way, and ii) to facilitate catchment selection based on the reliability of their attributes.

5. Ensure spatial consistency across the domain: We reduce the risk of generating artificial regional variations by using only data sets covering the entire CONUS, and not different data sets for different parts of the domain.



6. Select attributes directly relevant to hydrological modeling: The data sets we selected are commonly used for parameter estimation of hydrological and land surface models. A goal is to better assess how well those data sets capture the landscape features that matter for the storage and transfer of water across the landscape.

For most variables and catchments, the spatial resolution of the source data set (e.g., the remote-sensed vegetation characteristics) is smaller than the catchments, making upscaling necessary. By default, the upscaling was done using the arithmetic mean, except where indicated otherwise.

## 3. Location and topography

Location information and topographic indices were extracted for each catchment by N15 (Table 1). We display these
attributes on maps to introduce the main topographic features of the CONUS. Elevation obviously exerts a key control on catchment behavior (Figure 1a), as it strongly influences a wide range of catchment attributes that we present in this paper, such as soil depth, land cover, the fraction of the precipitation falling as snow or streamflow seasonality. Figure 1b illustrates that the eastern half of the CONUS is, with the exception of the Appalachian Mountains, much flatter than its western counterpart. Figure 1c shows the spatial distribution of catchment size and highlights that there are some large catchments:
five catchments have an area greater than 10,000km$^2$, and four of those are located in the in the Great Plains. Since we compute the catchment average of every attribute presented in this paper, it is important to keep mind that those catchment averages become less meaningful as the catchment area increases. In the context of hydrological modeling, the larger the catchment, the greater the need to account for spatial heterogeneity using some kind of spatially distributed representation.

As explained earlier, our aim is to reveal weaknesses in catchment characteristics and to discuss the impacts of such weaknesses for hydrological modeling. One way to do so is to compare different estimates of the same quantity, for instance catchment area. Two methods were used to determine the contours of each catchment, Geospatial Fabric (Viger, 2014; Viger and Bock, 2014) and GAGES II (Falcone, 2011). The polygons from Geospatial Fabric were instrumental to produce the N15 data set, since they were used to clip the gridded forcing data sets and the digital elevation model (from which elevation
bands were derived), and importantly, they were used to estimate the area of each catchment, which enabled the conversion of discharge at the catchment outlet to average runoff depth over the catchment. It is hence essential to determine if the area computed from the Geospatial Fabric polygon is reliable. We compared it to the area computed using the GAGES II data set and computed the absolute relative error between the two estimates. In eight catchments, the error is greater than 100% (red dots in Figure 1d), and in 62 catchments the relative error is greater than 10% (red and orange dots). Several of these
catchments are located in the Great Basin and in California where the Geospatial Fabric had difficulty identifying watershed boundaries. Additionally, the Geospatial Fabric was not designed to exactly replicate basin area above gauging locations, rather its development focused on continental scale hydrologic modeling, thus some basin area discrepancies are inherent in





the development of the Geospatial Fabric. We recommend not using catchments with large error discrepancies with GAGES-II, as they are most likely erroneous in the Geospatial Fabric (e.g., Bock et al., 2016). Note that in general, catchment delineation is more challenging in flat areas, but here errors in flat areas are relatively well contained, except in Florida.

## 4 Climate indices

### 4.1 Data and methods

Climate indices were derived using meteorological forcing data from N15. N15 includes forcing from three data sets (NLDAS, Maurer and DAYMET) but for the computation of the indices, only DAYMET data were used. All the climate indices and hydrological signatures (Section 5) were computed for the period 1989/10/01-2009/09/30 (hydrological years 1990 to 2009). The choice of this period was based on the proportion of missing daily discharge measurements (the forcing time series were all extracted from gridded data sets and are all complete). We consider this period to be long enough to derive climatological indices (in particular when rare events are characterized) and short enough to be little impacted by the lack of daily discharge measurements at the beginning and end of the period covered in N15 (1980-2014, see Figure 2).

There is a wide range of climatic indices in the literature. We selected indices with the goal to synthetize this myriad of possibilities and to provide direct support to the study of hydrological processes (Table 2). These indices characterize dry periods, high precipitation events and the baseline over two time scales: the daily time scale (e.g. frequency of high precipitation events) and the seasonal time scale (e.g. the proportion of precipitation falling as snow).

At the seasonal time scale, we computed three indices: aridity, the fraction of the precipitation falling as snow, and the seasonality and timing of precipitation. These three indices were previously used for the classification of 321 catchments across the CONUS and were shown to provide relevant insights into the relationship between catchment behavior and their physiographic characteristics (Berghuijs et al., 2014, note that we use slightly different formulations of these indices, see Table 2). Aridity is defined as the ratio of mean annual potential evapotranspiration over the mean annual precipitation. The occurrence of snow was estimated for daily time steps using a temperature threshold of 0°C. The seasonality and timing of precipitation are combined into a single metric, which relies on sine curves representing the annual cycle of precipitation and temperature. These three seasonal indices provide a good overview of the mean and seasonal climatic conditions, but do not explicitly consider dry periods and intense precipitation events, which occur at different time scales and are key drivers of droughts and floods. To fill this gap, we considered the frequency of dry days and high precipitation events, as well as the mean duration of these events and determined the season during which most of the high precipitation events and dry days occur. This provides some insights into the precipitation regime (convective or stratiform) and phase (liquid or snow).



## 4.2 Results and discussion

The annual precipitation cycle is strongest over the Pacific Coast (maximum in winter), over the northern Great Plains and Florida (maximum in summer) and weakest along the Atlantic Coast (Figure 3a). The fraction of precipitation falling as snow is highest over the Rocky, Cascade and Sierra Nevada mountain ragnes, followed by the Northeast and the Great Lakes

Region (Figure 3b). Aridity is the highest over the Southwest, High Plains and Great Plains, when in contrast, the Northwest, Northeast and the Appalachians are the most humid regions (Figure 3c). High precipitation events occur most frequently in winter along the Pacific Coast (Figure 3f) and are relatively long-lasting (Figure 3e), which reflects their large (synoptic) scale nature. In contrast, summertime convective systems (e.g., mesoscale systems) over the High Plains, Great Plains and the Upper and Middle Mississippi Valley generate the most frequent high precipitation events. In the band stretching from

Louisiana to Georgia, high precipitation events are most frequent in winter, as the result of the intense extratropical cyclone activity. The frequency of dry days (Figure 3g) is closely related to aridity (Figure 3c). Catchments located in the region stretching from California to Texas typically experience the longest periods of successive dry days, while those in the Northeast are at the other end of the spectrum (Figure 3h). Dry days are particularly frequent in summer west of the Rocky Mountains, in winter in the Great Plains and Mississippi Valley and in autumn in the Atlantic Coast States (Figure 3i).

## 5. Hydrological signatures

### 5.1 Data and methods

Hydrological signatures were chosen using a similar rationale as for climate indices: we aimed to capture the hydrological baseline, as well as low flow and high flow events. All signatures were computed using daily discharge time series retrieved by N15 from the USGS for the period 1989/10/01-2009/09/30 (Figure 2).

We selected signatures from the set that Sawicz et al. (2011) used to explore the similarity between 280 catchments in the Eastern USA and classify them (Table 3). The runoff ratio indicates how much of the long-term precipitation leaves the catchment as streamflow, thereby reflecting evaporation losses. We use the slope of the flow duration curve to characterize streamflow variability: steeper flow duration curves define greater variability over the year. The contribution of baseflow to

the total discharge is estimated by the baseflow index computed by hydrograph separation using a digital filter (Ladson et al., 2013). Hydrograph separation is often considered to be desperate (Hewlett and Hibbert, 1967; McDonnell, 2009; Beven, 2012) and it has to be recognized that the technique used for the separation influences the estimated baseflow index (e.g. Beck et al., 2013; Ladson et al., 2013). Hydrograph separation can nevertheless provides valuable insights into catchment behavior (e.g., Harman et al., 2011) and the baseflow index has proven to be a useful variable to compare and classify large

samples of catchments (e.g., Sawicz et al., 2011; Beck et al., 2016). Further, catchment response to a change in precipitation, which is in particular relevant in the context of climate change (e.g., Vano et al., 2015), was evaluated by computing the elasticity between annual precipitation and discharge. Finally, we characterized discharge seasonality using the date of the




center of mass. This indicator is frequently used to quantify the impacts of climate change on the hydrology of snow-dominated catchments (e.g., Court, 1962; Stewart et al., 2005; Addor et al., 2014). The center of mass dates have been shown to occur earlier, as temperature increases can force both an earlier onset of snow melt and a higher proportion of precipitation falling as rain.

Since the hydrological signatures introduced so far do not explicitly consider low and high flow events, we defined high and low flow days using thresholds based on the median and mean daily flow, respectively (Clausen and Biggs, 2000; Olden and Poff, 2003; Westerberg and McMillan, 2015). We computed the average duration and average frequency of high and low flow events. We also extracted $5^{th}$ and $95^{th}$ percentiles (Q5 and Q95, respectively) from the flow duration curve to characterize those events.

### 5.2 Results and discussion

The mean annual discharge and runoff ratio are strongly correlated (Figures 4a and b) and present clear similarities to catchment aridity (Figure 3c). In the Great Plains, where the evaporative demands exceed available precipitation (aridity > 1), more than 80% of the precipitation is evaporated (runoff ratio < 0.2) and the mean annual discharge is often as as low as 0.3 mm/day. In contrast, in the Pacific Northwest, precipitation is often twice as high as PET (aridity < 0.5). Both the runoff ratio and the mean annual discharge are higher in the Pacific Northwest than in the Northeast, as can be expected from the seasonality of precipitation, which peaks in winter in the Pacific Northwest (Figure 3a). Most of the discharge flows during the first half of the year in the Pacific Northwest (mean date of center of mass < 156, Figure 4c). Streamflow is in contrast delayed by snow accumulation in the Rocky Mountains, it is also late in the Midwest (in part because of the seasonality of the precipitation, which peaks in summer), and in contrast early in the band stretching from eastern Texas to South Carolina (which is consistent again with the seasonality of precipitation). Similarities exists between the patterns of the slope flow duration curve and the baseflow index (Figure 4d and e), with lower baseflow index and higher slopes both indicating flashy catchments, a clear example being the area stretching from east Kansas to Kentucky. Finally, at the annual scale, the discharge of more arid catchments tends to react more strongly to changes in precipitation (Figure 4f, see also Harman et al., 2011).

The frequency of low and high flow events is correlated (Figure 4g and j) and by definition, both frequencies are low in catchments with a low slope of the flow duration curve (Figure 4d). High flows are least frequent and the most short-lived in the Pacific Northwest and in the Appalachian Mountains, and when they occur their absolute discharge is higher than in other regions (Figure 4i). Q5 is more than ten times higher in the Pacific Northwest and in the Appalachian Mountains than in the most arid catchments, which reflects the capacity of these humid catchments to sustain baseflow.



Note that even though spatial patterns emerge from the maps in Figure 4, they tend to be less smooth than those of climate indices (Figure 3). In other words, there can be some strong variations over short distances, for instance in the slope of the duration curve or in signatures related to extreme (high and low) streamflow conditions. Plausible explanations are that i) hydrological signatures are the end result of the interactions between several non-linear processes (as opposed to the smaller

number of processes controlling for instance the fraction of precipitation falling as snow), ii) hydrological signatures are sensitive to uncertainties in discharge measurements (Mcmillan and Westerberg, 2015; Westerberg and McMillan, 2015), which we suspect generate noise (random variations) in the maps of particularly sensitive signatures, such as the slope of duration curve or signatures related to extreme streamflow conditions.

## 6. Soil characteristics

### 6.1 Data and methods

The soil characteristics we derived are principally based on the State Soil Geographic Database (STATSGO) data set post-processed by Miller and White (1998). Miller and White (1998) discretized the top 2.5m of soil into 11 layers, whose thickness increases with depth (from 5cm for the two top layers to 50cm for the three deepest ones). For each layer, they relied on the original STATSGO data to determine the dominant soil texture class. They considered a total of 16 classes, the

12 standard United States Department of Agriculture (USDA) soil texture classes plus four additional nonsoil classes characterized as organic material, water, bedrock and other.

We estimated the saturated hydraulic conductivity and porosity (saturated volumetric water content) of each layer using the multiple regressions relying on sand and clay fraction originally proposed by Cosby et al. (1984) and now commonly for

land surface modeling (e.g., Lawrence and Slater, 2008). For organic material, we used default values for the saturated hydraulic conductivity and porosity based on Lawrence and Slater (2008). Then for each SATSGO polygon, we computed the average of each soil characteristic (see list in Table 4) over the top 1.5m of soil using the following weighted mean:

$$X_p = {\sum_{i=1}^{i=9} X_i\,T_i} \Big/ {S_{depth}}$$

where $X_p$ designates the mean value of the variable $X$ over the 1.5m of soil (nine top layers, see points 1 and 2 below), $X_i$ is its value over layer $i$, $T_i$ is the thickness of layer $i$ and $S_{depth}$ is the cumulated depth of the layers. Then for each catchment, we computed the weighted mean of the soil characteristics of the STATSGO polygons within the catchment, the weight being the fraction of the catchment covered by each polygon. For hydraulic conductivity, the harmonic mean was used instead of the arithmetic mean, for the averaging along each soil column and across the catchment (see Samaniego et al. (2010) for a

discussion on upscaling operators).



Before we start interpreting the results of the aggregation of STATSGO data to the catchment scale, we consider important to discuss some key limitations of STATSGO. Those limitations were already underscored by Miller and White (1998) and also affect more recent soil data sets. It is our impression that although they reduce our ability to correctly reflect soil properties in hydrological models, they are commonly overlooked.

1.  Limited depth: Miller and White (1998) note that "[…] only about 2.5% of the components have layers extending below 203 cm (80 inches). Accordingly, the bottom two standard layers contain meaningful data only for a minority of the map units". In other words, although the STATSGO data set is often perceived as describing the top 2.5m of soil, over the majority of the CONUS only the top 1.5m are covered, and data from the bottom 1m in those areas are potentially misleading.

2.  Low information content in the deepest layers: They warn the reader that "for approximately half the components, the minimum and maximum depth-to-bedrock […] both have the value 152 cm (60 inches); in the great majority of these cases, this indicates that this was the maximum depth to which soil was normally examined and bedrock was not actually encountered". This means that when the two last layers (1.5 to 2.5m deep) are marked as bed rock, in about half of the cases, the bed rock has not actually been reached, which leads to an underestimation of the soil depth. Given these limitations, we decided to restrict our attention to the top 1.5m of soil (i.e., to the top 9 layers).

3.  Only fine fraction characterized: The STATSGO sand, clay and silt fraction are only for the portion of soil that is finer than 2mm. That is, STATSGO data should certainly not be considered to be representative of the whole soil column, but it is also important to keep in mind that it does not either completely characterize its top part, since only the soil fraction finer than 2mm is considered.

4.  Lack of representativeness of the dominant soil texture class: Miller and White (1998) stress that STATSGO "units may be quite internally heterogeneous, with as much as 50% of the map unit having soil properties that differ significantly from the map unit description".

5.  Scale inadequacy: although soil hydraulic properties can be measured in a lab, it is still unclear how to meaningfully upscale them to the catchment scale (these quantities can be characterized as incommensurate (e.g., Beven, 2012).

In a general sense, soil data sets only characterize the top soil layers, even when the soil can be much deeper (issues 1 and 2). In fact, the soil depth of a catchment indicated as 1.5m in STATSGO can be an order of magnitude greater. Uncertainties in soil depth are critical for hydrological modeling, in particular because they hamper the determination of the root zone storage capacity (Boer-Euser et al., 2016). To explore those uncertainties, we included a recently-released soil data set (Pelletier et al., 2016, refered to as P16 in continuation), from which we extracted the thickness of the permeable layers above bedrock, i.e. the depth to bedrock. The principal advantage of this data set is that it covers the top 50m of soil. It comes on a global 30 arcsec (~1 km) grid. We estimated the catchment average by computing the mean from all the grid points falling within each



catchment. It however does not provide information on soil texture classes, so it cannot be use to estimate variables like the saturated hydraulic conductivity or the porosity. Another key difference is that P16 leveraged geomorphological principles to obtain more precise estimates than what would be obtained by interpolating soil pit observations. We do not explicitly deal with problems 3 to 5 in this study, but expect them to cause lower than expected performance when hydrological modeling

relies on STATSGO or similar data sets.

## 6.2 Results and discussion

Once aggregated to the catchment scale, STATSGO data reveal the following features. Catchments with a sand fraction greater than 50% are predominantly located along the Gulf Coast and the Atlantic Coast, and in the Great Lakes Region (Figure 5a). This leads to a relatively low porosity fraction and high saturated hydraulic conductivity (Figure 5d and e).

Conversely, catchments with a silt fraction greater than 50% are mostly located in a band stretching from Kansas to New York (Figure 5b). Catchments in this band also tend to feature a comparatively large clay fraction (Figure 5c). This implies a higher than average porosity fraction (Figure 5d). Although variables like porosity and saturated hydraulic conductivity are commonly relied upon for parameter estimation, we note that their value in terms of process understanding should not be overestimated, given the limitations outlined in Section 6.1.

As for soil depth, STATSGO and P16both indicate that the soil is shallower in the Appalachian Mountains than along the Gulf and Atlantic Coasts. There are however disagreements in the Rocky Mountains (e.g., in Colorado) and in the Pacific Northwest: STATSGO indicates a soil depth equal or greater than 1.5m when the depth to bed rock according to P16 is smaller than 1m. The lack of quantitative agreement between the two data sets appears clearly in Figure 5i. The left hand

part of the figure (orange background) includes all the catchments in which there can potentially be an agreement between STATSGO and P16, since the estimated depth to bedrock is equal or smaller than 1.5m. There is however considerable scatter around the 1:1 orange curve, which illustrates the uncertainty in estimates of the soil depth, which directly impact estimates of maximum water content of soils (Figure 5f). This issue is even clearer when the right-hand side of Figure 5i is considered. In about half of the catchments (47%), the depth to bed rock is greater than 1.5m, so it cannot be covered by

STATSGO. For 24% of the catchments, the depth to bed rock estimated using P16 is greater than 15m, i.e., ten times the depth covered by STATSGO. This underscores the inability of data sets like STATSGO to provide a realistic characterization of soils in areas of high sedimentary deposition.

Finally, we considered three metrics that can be considered as metadata. The fraction of the catchment characterized as

"water" is relevant because it indicates the presence of lakes (Figure 5j). The "organic" fraction, which importantly impacts soil hydraulic properties, is typically negligible, but is non-negligible in many catchments in Florida and in the Great Lakes Region (Figure 5k). The fraction of soil marked as "other" (for which no soil characteristics are available, and which is ignored from the computation of all soil attributes) is significant in many catchments (Figure 5l). How detrimental that is





will depend on the application. One way to assess this effect when using soil characteristics to explain the performance of hydrological models, would be test whether a clearer relationship is obtained by progressively excluding catchments with the highest fraction of soil marked as "other".

## 7. Vegetation characteristics

### 7.1 Data and methods

We considered two key indicators of vegetation density: the leaf area index (LAI) and the green vegetation fraction (GVF), which approximates the vertical and horizontal density of vegetation, respectively. We used the 1km vegetation products derived from the Moderate Resolution Imaging Spectroradiometer (MODIS) data to estimate their climatological monthly values over 2002-2014. LAI is defined as the one-sided green leaf area per unit ground area in broadleaf canopies and as half

the total needle surface area per unit ground area in coniferous canopies. We extracted the maximum monthly LAI, which can be used to constrain the maximum evaporative capacity and vegetation interception capacity in models. Seasonal variations in LAI are principally related to trees growing and shedding their leaves. To quantify these variations, we computed the difference between the maximum and minimum monthly LAI. In absolute terms, these variations are highest in areas of deciduous broadleaf forest. The GVF can be used in models to estimate the proportion of each grid cell covered

by vegetation (1 minus the GVF gives the fraction of the grid cell from which evaporation occurs directly from the soil). Variations in the GVF are particularly high for croplands, as the result of the growing and harvesting of the crops. Like for the LAI, we extracted the maximum monthly value of the GVF and the difference between the maximum and minimum monthly value.

Additionally, we included the land cover class based on the International Geosphere-Biosphere Programme (IGBP) classification (Belward, 1996) derived from MODIS data. For each catchment, we defined the dominant land cover class as the most frequent class based on all the grid points fully or partially contained in the basin (each grid cell was weighted based on how much of it was contained within the basin boundaries). The fraction of the catchment covered by the dominant class is an indicator of the representativeness of the dominant class for the whole catchment.

Finally, based on the IGBP land cover class of each grid point, we approximated the root-depth distribution based on Zeng (2001). The distribution is estimated using a two-parameter equation, the value of these parameters being dependent on the IGBP land cover. The root fraction decreases exponentially with soil depth: the depth of the soil layer encompassing the top 50% of the root system is typically between 0.12 and 0.26m depending on the land cover, and for the top 99% of the root

system, this depth is typically between 1.4 and 2.4m and is often named "rooting depth". We computed the root-depth distribution for each grid point based on its land cover. We then extracted the values associated with the following





percentiles: 10, 25, 50, 75 and 99%. For each percentile, the catchment average was estimated using the arithmetic mean. Table 5 provides the complete list of vegetation attributes that we considered.

### 7.2 Results and discussion

The maximum LAI and GVF are highly correlated (Figure 6a and d, see also the mean value for each land cover class in Figure 6k), which reflects that short vegetation tends to be sparse and forests of taller trees tend to be dense, but could also indicate that the MODIS data used to compute these two fields do not enable us to fully differentiate between vertical and horizontal vegetation density. These two fields are similar to that of the fraction of forest (Figure 6c, positive correlation) and aridity (Figure 6c, negative correlation), with arid catchments typically associated with a lower LAI and lower GVF. Note that because the catchments selected are minimally impacted by human activities, none of them is classified as urban (Figure 6j).

The amplitude of the seasonal variations of LAI are strongly linked to LAI max (Figure 6a and b). Overall, catchments dominated by land cover classes with a high LAI (e.g., deciduous broadleaf forest or mixed forest, Figure 6j) tend to experience a significant increase and drop of LAI, a clear exception to this rule being evergreen broadleaf forests (Figure 6k). The seasonal variations of GVF are particularly high for croplands, which is expected and reflects the growing-harvesting cycle. However, note that there are also important seasonal changes in the catchment dominated by the deciduous broadleaf forests (Figure 6k), although the horizontal tree density does not change significantly. Again, this suggests that the MODIS data does not enable to fully differentiate between vertical and horizontal vegetation. Users might hence decide to consider LAI only to summarize seasonal land cover variations.

To explore spatial variations in rooting depth, we used two catchment attributes introduced earlier in this paper: aridity and depth to bedrock. Figure 6l shows that in water-limited catchments (aridity > 1) the rooting depth increases with aridity, which can be interpreted as a sign that trees increase their root zone storage capacity to compensate for the overall lack of water. In more humid catchments (aridity < 1), shallow soils seem to constrain the vertical development of roots of tall trees like evergreen needleaf forests and deciduous broadleaf, and in contrast, mixed forest and evergreen broadleaf forest can develop deeper roots. This simple example illustrates how the CAMELS data set enables the comparison of diverse attributes for a large number of catchments.

### 8. Comparison with the MOPEX data set

The CAMELS data set is similar to the data set produced for the Model Parameter Estimation Experiment (MOPEX; Duan et al., 2006; Schaake et al., 2006) in that it provides hydroclimatic time series and geophysical attributes for a large number of basins in the CONUS. MOPEX data have been used in large number of studies, including two catchment similarity studies



mentioned earlier (Sawicz et al., 2011; Berghuijs et al., 2014). For CAMELS, we use different criteria for catchment selection than those used for MOPEX, which leads to a relatively small overlap between the two data sets (they have 52 catchment are common, see Figure 7).

Both MOPEX and CAMELS require long observation time series and exclude catchments subject to human influence, but they use different approaches to characterize these aspects. For MOPEX, the stations part of the hydro-climatic data network (HCDN, Slack and Landwehr, 1992) together with those selected by Wallis et al. (1991) were considered to select potential basins. For CAMELS, an updated version of the HCDN classification was used (HCDN-2009, Lins, 2012): some catchments were excluded (e.g., because they no longer met the minimal disturbance criteria defined in the original HCDN report) and

other catchments were added (e.g., because their streamflow records, which were considered too short when the original HCDN was published, became long enough). It noteworthy that the number of reference stations went down from 1659 (HCDN) to 743 (HCDN-2009).

For a catchment to be part of the MOPEX data set, an essential criterion was that its number of rain gauges had to be higher

than a threshold based on the catchment area. This led to the exclusion of 77% of the potential MOPEX basins, resulting in only 438 basins considered to have a dense enough network of gauges. Although we do recognize the importance of reliable precipitation data for hydrological modeling, we did not exclude catchments based on their rain gage density. We argue that uncertainties in precipitation estimates (and in forcing in general) can now be assessed using independent data sets (e.g., Newman et al., 2015a, see also Section 9). We also consider that uncertainties in observed time series (in particular in

discharge records, see Coxon et al., 2015; Mcmillan and Westerberg, 2015) and uncertainties in catchment attributes (e.g. soil depth, see discussion in Section 6) can also lead to biased conclusions on hydrological processes, and hence should also be considered in the catchment selection processes. Yet the influence of these sources of uncertainties on research results will depend the catchments and variables of interest, so we leave it to the users to define their own criteria.

The differences between the catchment selection approaches followed for MOPEX and CAMELS have three main implications. First, CAMELS covers 671 catchments, whereas MOPEX covers 438 catchments. Second, CAMELS catchments are more evenly distributed across the CONUS and provide a better coverage of the Western half. Third, when the focus is on the last decades, catchments from HCDN-2009, on which CAMELS relies, are less likely to be influenced by human activities.

Finally, and maybe most importantly, we present a more detailed and transparent description of the origins and limitations of the data sets used to derive catchment attributes. A substantial part of this paper is dedicated to the discussion of limitations of the source data sets and we use competing approaches to estimate the same quantity, thereby revealing uncertainties in





those attributes. This is motivated by the belief that identifying weaknesses in catchment attributes helps us to anticipate how they might bias conclusions of hydrological studies.

**9. Online availability and possible future extensions**

The catchment attributes presented here will be in open access (freely available online) in spring 2017. Further details on
their format will be will be provided online. Please register online to be sure to receive updates when new variables will become available and when errors will be found and fixed.

Our intention with this data set is provide quantitative estimates of key geophysical attributes that shape catchment behavior. We see the data set in its current state as a starting point, and anticipate that it will be keep evolving and become more
exhaustive. In particular, there are two types of catchment attributes that are currently missing and that would be useful additions to the CAMELS data set:

1. Network characteristics, such as drainage density, catchment geometry and the distribution of stream orders have been shown to have key influences hydrological response (e.g., Rodríguez-Iturbe and Valdés, 1979).
2. Geological attributes are necessary to characterize the part of the catchment located below the soil layer described by STATSGO (top 1.5m) and to complete the information provided by P16 (top 50m). Characteristics such as geology permeability and porosity are essential to better understand and model subsurface water storage and transfer (e.g., McGuire and McDonnell, 2006).

Although it is our aim to enable users to assess the reliability of the attributes and to select catchments and interpret results accordingly, more work is necessary for a complete uncertainty assessment. Methods and data sets are however already available to quantitatively characterize uncertainties in following attributes:

1. Atmospheric forcing: the N15 data set provides forcing from three data sets (Daymet, NLDAS and Maurer) but in
this study we only use Daymet. Using the two other data sets might lead to some differences in climate indices, particularly when it comes to heavy precipitation events and/or to catchments with a sparse observation network. Another option to characterize the uncertainty in the forcing is to use the ensemble of gridded forcing produced by Newman et al. (2015a).
2. Discharge measurements: some hydrological signatures are more sensitive than others to uncertainties in discharge
measurements (Westerberg and McMillan, 2015). Methods to characterize those uncertainties and explore their propagation into hydrological signatures in a large sample of catchments exist (Coxon et al., 2015) but require detailed information on the rating curves used for discharge estimation, which were not readily available for this study.





3. Soils: the STATSGO data set is subject to several critical limitations, many of them being overcome by the recently released POLARIS data set (Chaney et al., 2016) and SoilGrids (Hengl et al., 2016). A key advantage of these two data sets is that they describe soil attributes in a probabilistic way, i.e., they rely on machine learning algorithms to deliver uncertainty estimates.

In summary, the CAMELS data set is the combination of two data sets, which are available for download separately:

1. Hydrometeorological time series from Newman et al. (2015b), with the data set DOI 10.5065/D6MW2F4D.
2. Catchment attributes introduced in this paper, with the data set DOI 10.5065/D6G73C3Q.

**10. Concluding remarks**

We introduced a new set of attributes for 671 catchments in the contiguous USA. Hydrometeorolgical time series for these catchments are provided by the Newman et al. (2015b) data set, and together with the attributes introduced here, constitute the CAMELS data set. The wide range of geophysical characteristics covered by these basins opens new opportunities to quantitatively explore how the interplay between climatic, hydrological, vegetation and soil attributes shapes catchment behavior. This enables us to test hypotheses and formulate conclusions valid in diverse conditions and not limited to a few

specific locations.

We produced a series of maps depicting catchment attributes over the contiguous USA. We used these maps to examine regional variations of a wide range of attributes and to illustrate the relationships between them. From a practical perspective, our synthesis of several data sources into a single data set at the catchment scale greatly simplifies the

comparative study of catchment characteristics and the exploration of their influence on hydrological processes.

An essential feature of this work is that it involves a critical assessment of the limitations of the data and methods used to derive catchment attributes, and a discussion of their consequences for process understanding and hydrological modeling. We highlight in particular uncertainties in soil attributes, and by reviewing the assumptions made during the production and

processing of the STATSGO data set, we aim to provide the context necessary to adequately manipulate and interpret these attributes. Other data sets also provide catchment characteristics for a large number of catchments, but deliver them in a deterministic way, without explicitly acknowledging their uncertainties.

Our intention with CAMELS was to start filling this gap. We plan to expand this data set by adding new catchment attributes

and refining our characterization of the uncertainties in catchment attributes, forcing and streamflow measurements. We designed the tables of this paper so that they fully describe the methods and data used to compute each attribute, in an effort to make our work transparent and reproducible.





To conclude, we envision that the CAMELS data set will enable progress on a wide range of hydrological challenges related to catchment similarity, model parameter estimation based on geophysical characteristics, model benchmarking, regional variations of model performance, and to the information content of geophysical data sets.

**Acknowledgements**

The National Center for Atmospheric Research (NCAR) is sponsored by the National Science Foundation. This study was supported by the US Army Corps of Engineers Climate Preparedness and Resilience programs. We thank Mike Barlage for his advice on vegetation characteristics. Data processing and visualization was performed with R (R Core Team, 2017). Most colors were chosen using ColorBrewer 2.0 (Brewer, 2017). To conclude, let us highlight that hydrology is all about the
storage and transfer of water across the landscape, and so are CAMELS.

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



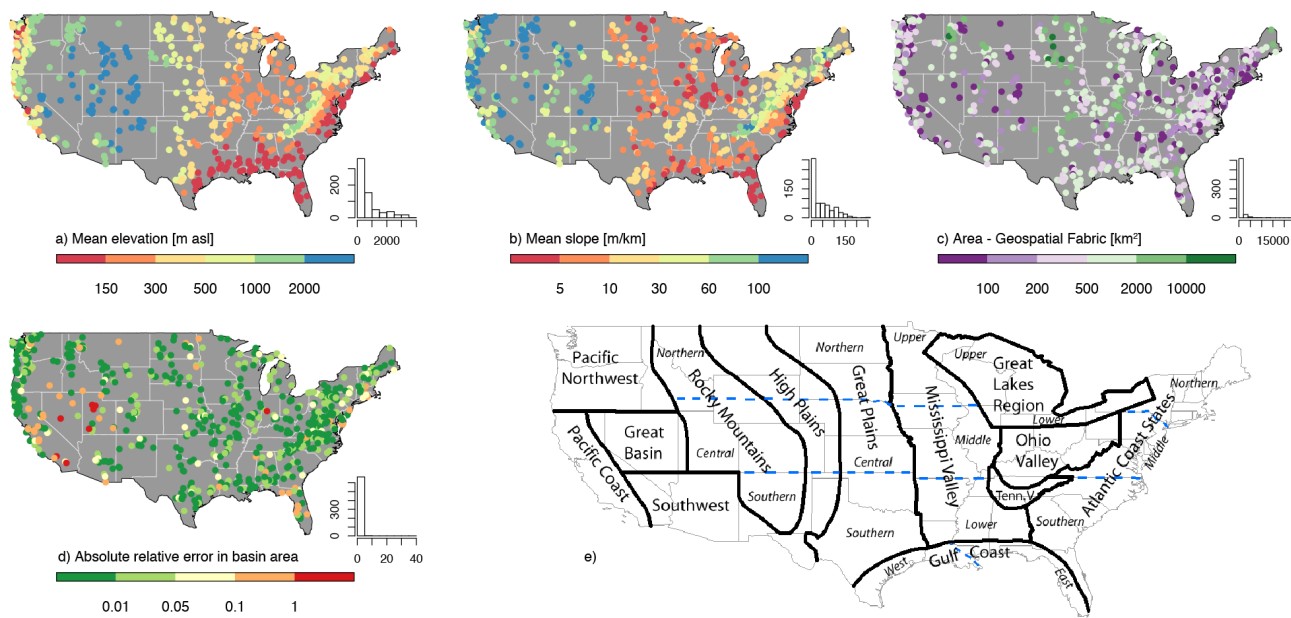

**Figure 1: a-d) Maps of topographic variables over the CONUS. The histograms indicate the number of catchments (out of 671) in each bin. e) Map of the regions referred to in this study (source: NOAA National Centers for environmental information, https://www.ncdc.noaa.gov/temp-and-precip/drought/nadm/geography)**





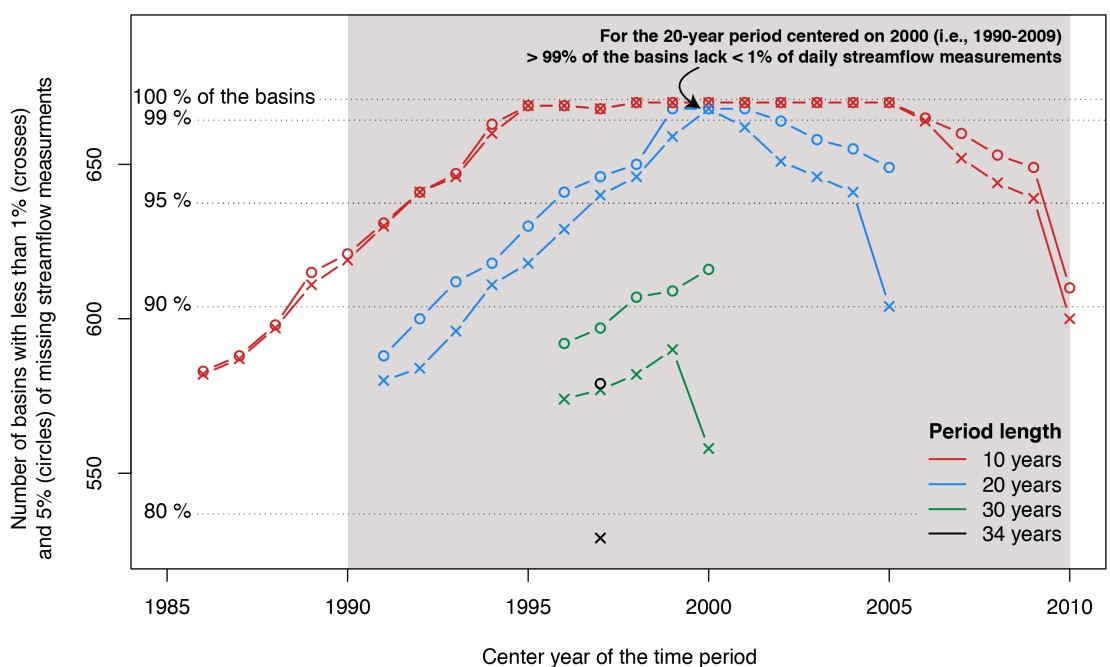

**Figure 2: Availability of streamflow measurements for periods of different length (colors) centered on different years (x-axis). The symbols (crosses and circles) indicate the number of catchments with at most 1% or 5% of daily streamflow measurements missing, respectively. The shape of the curves indicates that the proportion of missing data decreases from 1980 to 1990, stays low and then increases after 2010. Note that years are hydrological years (starting on October 1st).**







**Figure 3: Maps of the climate indices over the CONUS. The histograms and barplots indicate the number of catchments (out of 671) in each bin or category.**





**Figure 4: Maps of the hydrological signatures over the CONUS. The histograms indicate the number of catchments (out of 671) in each bin.**





**Figure 5: a-h and j-k) Maps of the soil characteristics over the CONUS. The histograms indicate the number of catchments (out of 671) in each bin. i) Comparison of the estimates of the depth to bed rock from STATSGO and P16. The orange area includes all the catchments for which there can be an agreement between the two data sets (i.e. estimates from both data sets are smaller than or equal to 1.5m). The orange curve is a 1:1 curve (note the logarithmic scale on the x-axis).**





**Figure 6: a) to h) Maps of the vegetation characteristics over the CONUS. The histograms indicate the number of catchments (out of 671) in each bin. i) Comparison of the LAI and GVF (maximum and difference between maximum and minimum) based on the dominant land cover class. j) Comparison of the mean aridity, mean total rooting depth and depth to bedrock for each land cover class, the color dots all have the same size.**





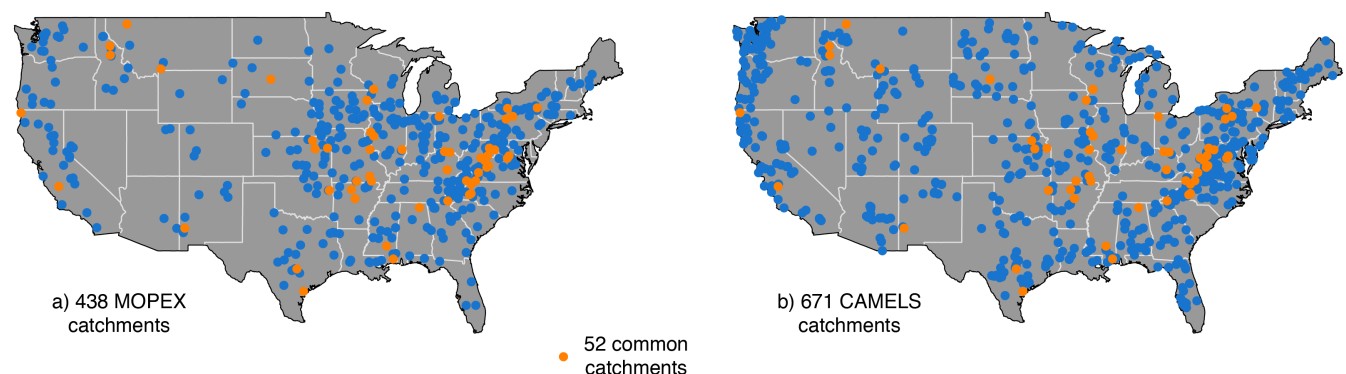

**Figure 7: A comparison of the spatial distribution of the catchments from a) the MOPEX data set and b) the CAMELS data set.**





**Table 1: Location and topography**

| Attribute | Description | Unit | Data source | References |
|---|---|---|---|---|
| gage_id | catchment identifier (8-digit USGS hydrologic unit code) | - | N15 - USGS data | - |
| huc_02 | region (2-digit USGS hydrologic unit code) | - | N15 - USGS data | - |
| gage_name | name of the gage, followed by the state | - | N15 - USGS data | - |
| gage_lat | latitude of the gage | ° north | N15 - USGS data | - |
| gage_lon | longitude of the gage | ° east | N15 - USGS data | - |
| mean_elev | mean elevation of the catchment | meter above sea level | N15 - USGS data | - |
| mean_slope | mean slope of the catchment | m/km | N15 - USGS data | - |
| area_gages2 | catchment area (GAGESII estimate) | km2 | N15 - USGS data | Falcone (2011) |
| area_geospa_fabric | catchment area (Geospatial Fabric estimate) | km2 | N15 - Geospatial Fabric | Viger (2014), Viger and Bock (2014) |

**Table 2: Climate indices. *: Computed over the period 1989/10/01 to 2009/09/30**

| Attribute | Description | Unit | Data source | References |
|---|---|---|---|---|
| aridity | aridity (PET/P, ratio of long-term average PET [estimated by *n2015* using Priestley-Taylor formulation] to long-term average precipitation) | - | N15 - Daymet* | - |
| precip_seas | seasonality and timing of precipitation (estimated using sine curves to represent the annual temperature and preciptiation cycles) | - | N15 - Daymet* | Eq. 14 in Woods et al. (2009) |
| frac_snow_daily | fraction of precipitation falling as snow (i.e. on days colder than 0°C based on daily precipitation and temperature values) | - | N15 - Daymet* | - |
| high_prec_freq | frequency of high precipitation days ( >= 5 times mean daily precip.) | days/year | N15 - Daymet* | - |
| high_prec_dur | average duration of high precipitation events (number of consecutive days >= 5 times mean daily precip.) | days | N15 - Daymet* | - |
| high_prec_timing | season during which most high precipitation days ( >= 5 times mean daily precip.) occur | - | N15 - Daymet* | - |
| low_prec_freq | average frequency of dry days ( <1 mm/day) | days/year | N15 - Daymet* | - |
| low_prec_dur | average duration of dry periods (number of consecutive days <1 mm/day) | days | N15 - Daymet* | - |
| low_prec_timing | season during which most dry days ( <1 mm/day) occur | - | N15 - Daymet* | - |





**Table 3: Hydrological signatures. *: Computed over the period 1989/10/01 to 2009/09/30**

| Attribute | Description | Unit | Data source | References |
|---|---|---|---|---|
| q_mean | mean discharge | mm/day | N15 - USGS data* | - |
| runoff_ratio | runoff ratio (Q/P, ratio of long-term average streamflow to long-term average precipitation) | - | N15 - USGS data* | Eq. 2 in Sawicz et al. (2011) |
| slope_fdc | slope of the flow duration curve (between the 33rd and 66th percentiles in log space) | | N15 - USGS data* | Eq. 3 in Sawicz et al. (2011) |
| baseflow_index | baseflow index (B/Q, ratio of long-term average baseflow to long-term average streamflow, hydrograph separation performed using Ladson et al. [2013] digital filter) | - | N15 - USGS data* | Eq. 4 in Sawicz et al. (2011), Ladson et al. (2013) |
| stream_elas | streamflow elasticity (sensitivity of streamflow to changes in precipitation at the annual time scale, using the long-term average as reference) | - | N15 - USGS data* | Eq. 7 in Sankarasubramanian et al. (2011), the last element being P/Q not Q/P |
| q5 | low flow percentile | mm/day | N15 - USGS data* | - |
| q95 | high flow percentile | mm/day | N15 - USGS data* | - |
| high_q_freq | average frequency of high flow days ( > 9 times the median daily flow) | days/year | N15 - USGS data* | Clausen and Biggs (2000), Table 2 in Westerberg and McMillan (2015) |
| high_q_dur | average duration of high flow events (number of consecutive days > 9 times the median daily flow) | days | N15 - USGS data* | Clausen and Biggs (2000), Table 2 in Westerberg and McMillan (2015) |
| low_q_freq | average frequency of daily low-flow days ( < 0.2 times mean daily flow) | days/year | N15 - USGS data* | Olden and Poff (2003), Table 2 in Westerberg and McMillan (2015) |
| low_q_dur | average duration of daily flow events (number of consecutive days < 0.2 times the mean daily flow) | days | N15 - USGS data* | Olden and Poff (2003), Table 2 in Westerberg and McMillan (2015) |
| hfd_mean | mean half flow date (date on which the cumulative discharge since October 1st reaches half of the annual discharge) | day of year | N15 - USGS data* | Court (1962) |





**Table 4: Soil characteristics. \*: Only covers the top 1.5m.**

| Attribute | Description | Unit | Data source | References |
|---|---|---|---|---|
| soil_depth_pelletier | soil_depth (maximum 50m) | m | Pelletier et al. | - |
| soil_depth_statgso | soil_depth (maximum 1.5m, layers marked as water and bedrock were excluded) | m | Miller and White (1998) - STATSGO* | - |
| porosity | volumetric porosity (saturated volumetric water content estimated using a multiple linear regression based on sand and clay fraction for the layers marked as USDA soil texture class and a default value [0.9] for layers marked as organic material, layers marked as water, bedrock and "other" were excluded) | - | Miller and White (1998) - STATSGO* | Table 4 in Cosby et al. (1984), Lawrence and Slater (2008) |
| conductivity | saturated hydraulic conductivity (estimated using a multiple linear regression based on sand and clay fraction for the layers marked as USDA soil texture class and a default value [36cm/hr] for layers marked as organic material, layers marked as water, bedrock and "other" were excluded) | cm/hr | Miller and White (1998) - STATSGO* | Table 4 in Cosby et al. (1984), Lawrence and Slater (2008) |
| max_water_content | maximum water content (combination of porosity and soil_depth_statgso, layers marked as water, bedrock and "other" were excluded) | m | Miller and White (1998) - STATSGO* | - |
| sand_frac | sand fraction (of the soil material smaller than 2 mm, layers marked as oragnic material, water, bedrock and "other" were excluded) | % | Miller and White (1998) - STATSGO* | - |
| silt_frac | silt fraction (of the soil material smaller than 2 mm, layers marked as oragnic material, water, bedrock and "other" were excluded) | % | Miller and White (1998) - STATSGO* | - |
| clay_frac | clay fraction (of the soil material smaller than 2 mm, layers marked as oragnic material, water, bedrock and "other" were excluded) | % | Miller and White (1998) - STATSGO* | - |
| water_frac | fraction of the top 1.5m marked as water (class 14 ) | % | Miller and White (1998) - STATSGO* | - |
| organic_frac | fraction of soil_depth_statgso marked as organic material (class 13) | % | Miller and White (1998) - STATSGO* | - |
| other_frac | fraction of soil_depth_statgso marked as other (class 16) | % | Miller and White (1998) - STATSGO* | - |





**Table 5: Vegetation characteristics. \*: Computed over the period 2002 to 2014.**

| Attribute | Description | Unit | Data source | References |
|---|---|---|---|---|
| frac_forest | fraction of forest | - | N15 - USGS data | - |
| lai_max | maximum monthly mean of the leaf area index (based on 12 monthly means) | - | MODIS* | - |
| lai_diff | difference between the maximum and mimumum monthly mean of the leaf area index (based on 12 monthly means) | - | MODIS* | - |
| gvf_max | maximum monthly mean of the green vegetation fraction (based on 12 monthly means) | - | MODIS* | - |
| gvf_diff | difference between the maximum and mimumum monthly mean of the green vegetation fraction (based on 12 monthly means) | - | MODIS* | - |
| dom_soil_cover | dominant soil cover (Noah-modified 20-category IGBP-MODIS land cover) | - | MODIS* | - |
| frac_dom_soil_cover | fraction of the basin associated with the dominant soil cover | - | MODIS* | - |
| root_depth_XX | root depth (percentiles XX = 10, 25, 50, 75 and 99% extracted from a root depth distribution based on IGBP land cover) | m | MODIS* | Eq. 2 and Table 2 in Zeng (2011) |