# Peer review of "The CAMELS data set: catchment attributes and meteorology for large-sample studies"

_Hydrology and Earth System Sciences, 2017_

## Referee Comment (RC1) · Anonymous Referee #1 · 2 May 2017

This manuscript describes and presents a dataset of catchment and hydroclimatic attributes for a set of un-impacted (less impacted?) catchments in the continental USA to facilitate large-sample / comparative hydrologic research. The dataset presented in this paper is a significant contribution to large-sample hydrology and worthy of publication. When combined with the time series records provided by Newman et al. (2015), the CAMELS dataset will allow researchers around the world to quickly test a range of hypotheses without spending significant amounts of time (∼months) re-collating a similar dataset. Making the dataset freely available online is excellent and is an example for other researchers to follow. Overall the manuscript is very well written and presented and only requires very minor changes as detailed below.

Minor comments

[Figure]

Abstract: It would be good to indicate in the Abstract that the catchments are un-impacted / less impacted by anthropogenic changes. This important point is buried on Page 12, line 9.

P3, L3: change "data sets used to for their" to "data sets used for their".

P4, L15: remove repetition of "in the" before "Great Plains".

P5, L25: The seasonality and timing of precipitation and temperature are summarised by sine curves fit to the monthly mean values. The authors need to report on the goodness-of-fit of these sine curves as the resulting single metric is based on the sine curve's, which may, or may not, provide a good fit to the data.

P6, L2: Please provide an explanation of what + and - values of the seasonality metric (Figure 3a) actually mean.

P6, L4: "ragnes" should be "ranges".

P6, L26: "Hydrograph separation is often considered to be desperate" – although Beven (2012) makes a similar statement, it would be helpful to the reader to know the wider context of this statement. Why is hydrograph separation considered desperate?

P6, L28: change "provides" to "provide".

P7, L14: remove repetition of "as" before "low as".

P7, L21: change "slope flow" to "slope of the flow".

P8, L6 + other locations: change "Mcmillan" to "McMillan".

P9, L1: change "consider important" to "consider it important".

P10, L1: change "It however" to "However, it".

P10, L16: change "P16both" to "P16 both".

P12, L9: change "them is classified" to "them are classified".

P12, L31: change "used in large" to "used in a large".

P13, L3: change "catchment are common" to "catchments in common".

P13, L11: change "It noteworthy" to "It is noteworthy".

P13, L13: change "depend the catchments" to "depend on the catchments".

P14, L9: change "it will be keep" to "it will keep".

Reference to Ladson et al (2013): change "Bronw" to "Brown".

Figure 2 Caption: change "5%of daily" to "5% of daily"

Figure 5: To make the comparison between 5g and 5h easier to see, I recommend using a single scale for the two plots. In this way the same colour would mean the same soil depth in both maps. In the current version dark green means two different soil depths, which is confusing to the reader.

Figure 5 Caption: change "a-h and j-k" to "a-h and j-l"

Figure 6 Caption: change "a) to h)" to "a) to j)"

[Figure]

---

## Referee Comment (RC2) · Anonymous Referee #2 · 4 May 2017

This manuscript presents a nice extension of Newman et al., 2015b on a catchment dataset across the U.S. My group have used the Newman dataset in our research new process understanding, so I am happy to see the extension of it. I feel this manuscript can be published at HESS after addressing the following comments.

1. MOPEX dataset is a very good example of such a catchment dataset. It has been extensively used by the hydrology and land surface modeling communities leading to at least over 100 journal articles. It'd be interesting to see a more in-depth or more detailed comparison between MOPEX and the new dataset here, i.e., a table would be nice.

2. I'd like to see more (perhaps quantitative) discussion on whether and how the catchments included in this dataset are free of human impacts. One good example is Wang

and Hejazi, 2011.

Wang D. and M. Hejazi (2011), Quantifying the relative contribution of the climate and direct human impacts on mean annual streamflow in the contiguous United States, Water Resources Research, 47, W00J12, doi:10.1029/2010WR010283

---

## Author Comment (AC1) · 24 Jul 2017

**"The CAMELS data set: catchment attributes and meteorology for large-sample studies" *by* Nans Addor et al.**

**Response to Anonymous Referee #1 by Nans Addor et al.**

*We thank the reviewer for her/his helpful and positive comments. Before we address them, we would like to stress that since our manuscript was submitted, we added a new set of attributes to CAMELS. We extracted geological characteristics from the GLiM and GLHYMPS data sets and produced catchment-scale averages for the 671 catchments. We think that this addition enables a more complete description of the landscape of the CAMELS catchments, and that it will provide useful insights into hydrological processes. These new attributes are now introduced and discussed in Section 8, Figure 7 and Table 6 – see the end of this document.*

*We are also pleased to report that the catchments attributes are now freely available online: https://dx.doi.org/10.5065/D6G73C3Q*

This manuscript describes and presents a dataset of catchment and hydroclimatic attributes for a set of un-impacted (less impacted?) catchments in the continental USA to facilitate large-sample / comparative hydrologic research. The dataset presented in this paper is a significant contribution to large-sample hydrology and worthy of publication. When combined with the time series records provided by Newman et al. (2015), the CAMELS dataset will allow researchers around the world to quickly test a range of hypotheses without spending significant amounts of time (~months) re-collating a similar dataset. Making the dataset freely available online is excellent and is an example for other researchers to follow. Overall the manuscript is very well written and presented and only requires very minor changes as detailed below.

Minor comments

Abstract: It would be good to indicate in the Abstract that the catchments are un- impacted / less impacted by anthropogenic changes. This important point is buried on Page 12, line 9.

*Thank you for the suggestion. We now mention in the first sentence of the abstract that the catchments are minimally impacted by human activities:*

*"We present a new data set of attributes for 671 catchments in the contiguous USA (CONUS) minimally impacted by human activities."*

*We now also stress this on P2, L27-30, and provide a reference to the section of the N15 paper*
*explaining how the catchments were selected, and in particular, how the impacts of human*
*activities were assessed:*

*"All those catchments have 20 years of continuous discharge record from 1990 to 2009 and are*
*minimally impacted by human activities (see Section 2.1 in Newman et al., 2015)."*

P3, L3: change "data sets used to for their" to "data sets used for their".

*Changed.*

P4, L15: remove repetition of "in the" before "Great Plains".

*Removed.*

P5, L25: The seasonality and timing of precipitation and temperature are summarised by sine
curves fit to the monthly mean values. The authors need to report on the goodness-of-fit of these
sine curves as the resulting single metric is based on the sine curve's, which may, or may not,
provide a good fit to the data.

*We added the following text to P5, L26-29:*

*"Note that sine curves do not necessarily provide a good fit to the annual precipitation cycle, for*
*instance in areas experiencing a strong annual cycle and multiple consecutive months with low*
*precipitation, such as California (see Berghuijs and Woods, 2015, for a solution to this issue),*
*yet they enable a first-order characterization of the dominant climatological features of diverse*
*locations, which is useful for studies such as this one."*

*Berghuijs, W. R., & Woods, R. A. (2015). A simple framework to quantitatively describe monthly*
*precipitation and temperature climatology. International Journal of Climatology, 36(9),*
*3161–3174. https://doi.org/10.1002/joc.4544*

P6, L2: Please provide an explanation of what + and - values of the seasonality metric (Figure
3a) actually mean.

*We added the following to Table 2: "positive [negative] values indicate that precipitation peaks*
*in summer [winter], values close to 0 indicate uniform precipitation throughout the year"*

P6, L4: "ragnes" should be "ranges".

*Changed.*

P6, L26: "Hydrograph separation is often considered to be desperate" – although Beven (2012)
makes a similar statement, it would be helpful to the reader to know the wider context of this
statement. Why is hydrograph separation considered desperate?

*We rephrased this sentence to avoid confusion:*

*"It has to be recognized that the technique used for the separation influences the estimated*
*baseflow index (see Beck et al., 2013; Ladson et al., 2013 for recent examples), yet hydrograph*
*separation can provide valuable insights into catchment behavior (e.g., Harman et al., 2011) and*
*the base flow index has proven to be a useful variable to compare and classify large samples of*
*catchments (e.g., Sawicz et al., 2011; Beck et al., 2016)."*

P6, L28: change "provides" to "provide".

*Changed.*

P7, L14: remove repetition of "as" before "low as".

*Removed.*

P7, L21: change "slope flow" to "slope of the flow".

*Changed.*

P8, L6 + other locations: change "Mcmillan" to "McMillan".

*Changed.*

P9, L1: change "consider important" to "consider it important".

*Changed.*

P10, L1: change "It however" to "However, it".

*Changed.*

P10, L16: change "P16both" to "P16 both".

*Changed.*

P12, L9: change "them is classified" to "them are classified".

*Changed.*

P12, L31: change "used in large" to "used in a large".

*Changed.*

P13, L3: change "catchment are common" to "catchments in common".

*Changed.*

P13, L11: change "It noteworthy" to "It is noteworthy".

*Changed.*

P13, L13: change "depend the catchments" to "depend on the catchments".

*Changed.*

P14, L9: change "it will be keep" to "it will keep".

*Changed.*

Reference to Ladson et al (2013): change "Bronw" to "Brown".

*Changed.*

Figure 2 Caption: change "5%of daily" to "5% of daily"

*Changed.*

Figure 5: To make the comparison between 5g and 5h easier to see, I recommend using a single
scale for the two plots. In this way the same colour would mean the same soil depth in both
maps. In the current version dark green means two different soil depths, which is confusing to
the reader.

*We agree that using different scales might be confusing at first, yet because the two data sets
cover very different ranges (0 to 1.5m for STATSGO versus 0 to 50m for Pelletier et al., 2016),
using the same scale leads to a significant loss of details in the one of the two maps. Our
intention with these two maps is to provide an overview of the spatial variations over the
CONUS. Figure 5i allows in contrast for a direct and more quantitative comparison of the two
data sets. We hence decided to maintain the scales used for Figures 5g and 5h.*

Figure 5 Caption: change "a-h and j-k" to "a-h and j-l"

*Changed.*

Figure 6 Caption: change "a) to h)" to "a) to j)"

*Changed.*

*Geological characteristics – new section and associated table and figure*

**8   Geology**

**8.1 Data and methods**

[revised manuscript text omitted]

**New references**

Gleeson, T., Moosdorf, N., Hartmann, J., & van Beek, L. P. H. (2014). A glimpse beneath earth's surface: GLobal HYdrogeology MaPS (GLHYMPS) of permeability and porosity. *Geophysical Research Letters*, *41*, 3891–3898. https://doi.org/10.1002/2014GL059856

Hartmann, J., & Moosdorf, N. (2012). The new global lithological map database GLiM: A representation of rock properties at the Earth surface. *Geochemistry, Geophysics, Geosystems*, *13*(12), 1–37. https://doi.org/10.1029/2012GC004370

[Figure]

Figure 7: Maps of the geological characteristics over the CONUS. The histograms indicate the number of catchments (out of 671) in each bin.

**Table 6: Geological characteristics.**

| Attribute | Description | Unit | Data source | References |
|---|---|---|---|---|
| glim_1st_class | most common geologic class in the catchment | qualitative | GLiM | Hartmann and Moosdorf (2012) |
| glim_1st_frac | fraction of the catchment area associated with its most common geologic class | - | GLiM | Hartmann and Moosdorf (2012) |
| glim_2nd_class | 2nd most common geologic class in the catchment | qualitative | GLiM | Hartmann and Moosdorf (2012) |
| glim_2nd_frac | fraction of the catchment area associated with its 2nd most common geologic class | - | GLiM | Hartmann and Moosdorf (2012) |
| glim_carb_rocks_frac | fraction of the catchment area characterized as "Carbonate sedimentary rocks" | - | GLiM | Hartmann and Moosdorf (2012) |
| glhymps_porosity | subsurface porosity | - | GLHYMPS | Gleeson et al. (2014) |
| glhymps_permeability | subsurface permeability (log10) | m2 | GLHYMPS | Gleeson et al. (2014) |

---

## Author Comment (AC2) · 24 Jul 2017

**"The CAMELS data set: catchment attributes and meteorology for large-sample studies" *by* Nans Addor et al.**

**Response to Anonymous Referee #2 by Nans Addor et al.**

*We thank the reviewer for her/his helpful and positive comments. Before we address them, we would like to stress that since our manuscript was submitted, we added a new set of attributes to CAMELS. We extracted geological characteristics from the GLiM and GLHYMPS data sets and produced catchment-scale averages for the 671 catchments. We think that this addition enables a more complete description of the landscape of the CAMELS catchments, and that it will provide useful insights into hydrological processes. These new attributes are now introduced and discussed in the new Section 8, Figure 7 and Table 6 – see the end of this document.*

*We are also pleased to report that the catchments attributes are now freely available online: https://dx.doi.org/10.5065/D6G73C3Q*

This manuscript presents a nice extension of Newman et al., 2015b on a catchment dataset across the U.S. My group have used the Newman dataset in our research new process understanding, so I am happy to see the extension of it. I feel this manuscript can be published at HESS after addressing the following comments.

1. MOPEX dataset is a very good example of such a catchment dataset. It has been extensively used by the hydrology and land surface modeling communities leading to at least over 100 journal articles. It'd be interesting to see a more in-depth or more detailed comparison between MOPEX and the new dataset here, i.e., a table would be nice.

*We rephrased and extended the end of Section 9. Comparison with the MOPEX data set (previously Section 8):*

*"Overall, the data used for CAMELS are more recent than those used for MOPEX. The period covered by hydro-meteorological times series is 1948-2003 for MOPEX and 1980-2015 for CAMELS, so given the fast rate of human development and the impacts caused by climate change, CAMELS provides a more current picture of hydrological processes in the United States. Further, CAMELS leverages new data sets, which were not available when the MOPEX data were released, for instance to characterize soils (Pelletier et al., 2016) and geology: GLiM (Hartmann and Moosdorf, 2012) and GLHYMPS (Glesson and et al., 2014). And importantly, data used for CAMELS are not only more recent, but also tend to be better documented. A clear example is that CAMELS meteorological time series come from three widely-used gridded data sets (Daymet, Maurer and NLDAS), while for MOPEX, station measurements were aggregated to provide catchment-scale estimates (Schaake et al, 2006)".*

*We also produced a new table outlining the main differences and similarities between*
*MOPEX and CAMELS (now Table 8):*

Table 8: Main differences and similarities between MOPEX and CAMELS

| | MOPEX | CAMELS |
|---|---|---|
| Number of catchments and spatial repartition | 438 catchments, principally from the Eastern half of the CONUS and with an under-representation of the Rocky Mountains | 671 catchments, with a relatively even repartition over the CONUS |
| Catchments in common to MOPEX and CAMELS | 52 | |
| Period covered by the hydro-meteorological time series | 1948 - 2003 | 1980 - 2015 |
| Minimal anthropogenic influence and long streamflow records assessed by: | Wallis et al. (1991) and the hydro-climatic data network (HCDN, Slack and Landwehr, 1992) | Updated version of the HCDN (HCDN-2009, Lins, 2012) |
| Raingauge density used as catchment selection criterion | Yes | No |
| Meteorological data | Point observations from National Climate Data Center daily Cooperative Observer Network (COOP) and SNOTEL stations. Missing observations were filled. Long-term precipitation averages computed using 1961-1990 PRISM data | Gridded data from Daymet (Thornton et al., 2012), Maurer (Maurer et al., 2002) and NLDAS (Xia et al., 2012) |
| Streamflow data | USGS streamflow measurements | |
| Soil data | SATASGO (Miller and White, 1998) | STATSGO (Miller and White, 1998) and Pelletier et al. (2016) |
| Vegetation data | North America Land Data Assimilation (NLDAS) | MODIS imagery |
| Geology data | Not available | GLiM (Hartmann and Moosdorf, 2012) and GLHYMPS (Glesson and et al., 2014) |
| Reference papers | Duan et al. (2006) and Schaake et al. (2006) | Newman et al. (2015) and this paper |

2. I'd like to see more (perhaps quantitative) discussion on whether and how the
catchments included in this dataset are free of human impacts. One good example is Wang
and Hejazi, 2011.

*We now mention in the first sentence of the abstract that the catchments are minimally*
*impacted by human activities:*

*"We present a new data set of attributes for 671 catchments in the contiguous USA (CONUS)*
*minimally impacted by human activities."*

*We now also stress this on P2, L27-30, and provide a reference to the section of the N15*
*paper explaining how the catchments were selected, and in particular, how the impacts of*
*human activities were assessed:*

[revised manuscript text omitted]

---

## Author Response (AR1)

[revised manuscript text omitted]